# A Review on LiFi Network Research: Open Issues, Applications and Future Directions

Rozin Badeel [ID], Shamala K. Subramaniam *, Zurina Mohd Hanapi [ID] and Abdullah Muhammed [ID]

Department of Communication Technology and Network, University Putra Malaysia (UPM), Seri Kembangan 43300, Selangor, Malaysia; rozinbabdal1987@gmail.com (R.B.); zurinamh@upm.edu.my (Z.M.H.); abdullah@upm.edu.my (A.M.)
* Correspondence: shamala_ks@upm.edu.my

**Abstract:** This paper extensively reviews and analyses Light Fidelity (LiFi), a new technology that uses light to transmit data as a high-speed wireless connection system from a wide spectrum of domains. An in-depth analysis and classifications of pertinent research areas for LiFi networks are presented in this paper. The various aspects constituting this paper include a detailed literature review, proposed classifications, and statistics, which further is deliberated to encompass applications, system architecture, system components, advantages, and disadvantages. LiFi and other technologies are compared, multi-user access techniques used in LiFi networks are investigated and open issues are addressed in detail. The paper is concluded with a comprehensive taxonomy of literature comparison that has served as the basis of the proposed open issues and research trends.

**Keywords:** hybrid wireless networks; LiFi; multiple access

## 1. Introduction

With the rapid growth of digital technologies, especially those in the networking and communication domains, much research has been conducted to prove the possibility of using Visible Light (VL) as a wireless medium for transferring data between smartphones, tablets, laptops and other mobile devices with respect to the 5th generation (5G) technology [1–4]. As a result, Harald Haas, a German physicist, presented a new technology called Light Fidelity (LiFi) [5]. LiFi is also a word used to describe high-speed networks that transport data using visible light. LiFi is a data transmission method that uses Light Emitting Diodes (LEDs) to send light signals. A photodiode device attached to the device receives these signals and provides access to the data, such as photos, videos, documents, or the Internet. LiFi is a complementary technology to Wi-Fi that relieves congestion in the radio spectrum while providing Internet access to the public. LiFi is a form of Optical Wireless Communication (OWC) that has no side effect on the human well-being due to the fact it is a wireless technology that utilizes VL as a communication medium instead of the basic radio wave, which relies on the electromagnetic spectrum [6]. The motivation behind using LiFi in many research areas, like hospitals and the medical sector, is largely attributed to the fact that it does not have interference that generally comes from other devices that use Radio Frequency (RF) which subsequently may lead to signal lost [7]. Therefore, LiFi is considered amongst the best alternatives to Wi-Fi in hospitals as it transcends the interference signal, which may occur from mobile and pc, leading to signal loss [7]. Therefore, VL-based technology such as LiFi is used increasingly in places where medical applications exist, such as Magnetic, which represent the overall bandwidth of the optical spectrum [8]. These are several hundred Terahertz (THz), which is much broader than the entire RF spectrum. Although IR is part of the electromagnetic spectrum that is not noticeable by human eyes, the radiation can also be called infrared light [9].

On the other hand, VL is visible and detectable by the human eyes. Furthermore, the spectrum of VL has seven colors [10], with their respective wave lengths and data

transmission speeds. Many studies have used and tested different lights to transmit data. The main focus was on Red, Green and Blue, which is also known as RGB [2].

One of the most important events that has made history in the 1880s was the invention of the photophone by Alexander Graham Bell and his team [11]. The photophone was an instrument that conveyed speech wirelessly via a beam of light. The first documented research in modern VLC began at the Keio University in Japan in Nakagawa Laboratory [12]. In 2006, CICTR researchers from the State University of Pennsylvania suggested the combination of power transmission with LED technology to ensure adequate broadband access. This study specifically focused on data transfer that utilized LEDs with VL. This channel provided inexpensive and sufficient illumination and home networking. Two years later, the European Union started its VLC research project, known as OMEGA [13]; they aimed to develop a high-speed domestic service network at rates of up to 1 Gbit/s. The maximum performance cap for the network was 1.25 Gbit/s. In 2008, the United States National Science Foundation conducted a range of tests to enhance the usage of smart lighting wireless communication network [13]. Funding was initiated to use LED bulbs as comparable points of communication technologies. Many other research ventures have been conducted in the European Union and the United States. Some of these works have contributed substantially to the establishment of LiFi at the Edinburgh University in Scotland. The development of the VLC technology began in 2006 and was primarily based on the two-way data communication. Subsequently, paving the way for the growth of LiFi technology. LiFi was introduced to the world via the Hass' 2011 TED talk. Wireless data transmission by light bulb was demonstrated in the talk. During the talk, Haas presented a variant of "The Light Fidelity". In 2012, he helped assess the LiFi Product Marketing Business, the first company to combine LiFi instruments with existing LED lights and named it Market-Built Equipment Original Equipment Manufacturer (OEM) [13].

One of the controversial concepts about LiFi and VLC is their common characteristics where many studies have stated and treated them as one, while others assume, they are different. Given the fact both use visible light as a medium of communication, it is important to point out the features and various components within different approaches and applications that distinguish LiFi from VLC. Specifically, LiFi is recognized as one of the most important technologies for 5G [14]. It is based on the VLC approach, which uses LEDs that are extensively used in houses, workplaces, and road systems to enable high-speed wireless communication. The transmission rates of over 3 Gb/s may be achieved utilizing Orthogonal Frequency Division Multiplexing (OFDM) [15]. However, unlike VLC, where the primary goal is to create a point-to-point link between different devices using visible light, LiFi is a light-based data transmission system that allows for fully interconnected cellular infrastructure, including bi-directional multi-user communication. Due to VLC's restricted transmission range, LiFi leads to further reduced cell size, which raises the size of the service area attocell [15].

LiFi is a high-speed wireless connection system [16] that requires multiple access points (APs) that form dense optical attocellular networks [4]. Inside an indoor area, data transmission through LiFi is indeed safer. The number of users who can connect to a LiFi AP is determined by the size of the attocell. LiFi signals appear once the AP's LED has been illuminated. The signals don't pass through walls. However, it is more challenging in the outdoor area because unauthorized users are more likely to detect LiFi transmitted signals without many constraints. Normally, the light bulb glows at a constant current supply, and a fast and stable electric current is required to generate stable optical outputs. LiFi only requires light, therefore it can be easily applied to any field where RF interaction is often difficult [17].

LED lights are used in LiFi to deliver good lighting since it uses the existing lighting infrastructure and provides illumination in indoor areas. In order to guarantee the general indoor illumination, a pre-requisite requirement of a minimum illuminance of 500 lux is needed, and the area optical power of LiFi APs is set to be 2.5 W/m2 [18]. Note that the measuring unit for illumination is lux [5,10,19–27]. LiFi technology has attracted high attention of the research community [6], and various research efforts on this technology have constantly been growing [6]. This technology is best known for its various importance and contributions over other existing technologies (i.e., Wi-Fi, Bluetooth, and RF identification) [6], which include high data transmission rates [9], fast speed [26], safety [5], availability [4], efficiency [28], security [29] and low cost [30]. The integration of LiFi in every light source creates an opportunity to shift ubiquitous light to the connected objects. Therefore, boosting its potentials and expansion in tremendous ways [6]. Extensive research on this technology and its various potentials will yield remarkable findings from the perspectives of science and academia and from various interested parties that have the commercial ability to integrate this technology in their respective domains. Organizations also will benefit from the full potential, and it will pave the way for new communications in the future. Thus, it is a very important topic that receives considerable attention that has served as an important motivating factor for this paper. This paper provides a comprehensive review of this technology from various aspects, and it addresses its unique features, research trends, and open issues to be pursued to further contribute towards the empowerment and deployment of this technology.

The contributions of this study can be summarized as follows: (I) a detailed literature review and a multi-aspect classification is introduced. (II) Identification of the most important features of the architectural component of LiFi network systems. (III) Detailed discussion on the advantages and disadvantages of this technology. (IV) A comprehensive comparison between LiFi and other technologies. (V) discussion of multi-user access in LiFi networks. (VI) discussions on the most significant scientific challenges and open issues for LiFi and finally a comprehensive discussion of the highly utilized area of LiFi applications.

This paper is organized into ten sections. Section 2 describes the review protocol used for the respective research methods, literature sources, and filtering steps in selecting research articles. In Section 3, the classification and statistical analysis results from the final set of reviewed articles are also outlined. Section 4 presents LiFi applications. Section 5 presents the components and architecture of the LiFi system. Section 6 discusses the advantages and disadvantages of LiFi in detail. Section 7 compares the main differences between LiFi and Wi-Fi. Section 8 explains the types of multi-user access used in LiFi. Section 9 presents the open issues and discussion. Section 10 presents the limitations and new future research directions. Section 10 concludes this study.

## 2. Review Protocol

This section discusses the main settings applied in searching, downloading, and filtering the papers included in this study. This section contains the details for the methods used followed by data sources in which the articles are searched, downloaded, and then filtered. Other significant elements, including the techniques on the manner studies have been selected through the eligibility criteria, are included in this section. Finally, the data collection process and its respective literature are discussed in Figure 1.

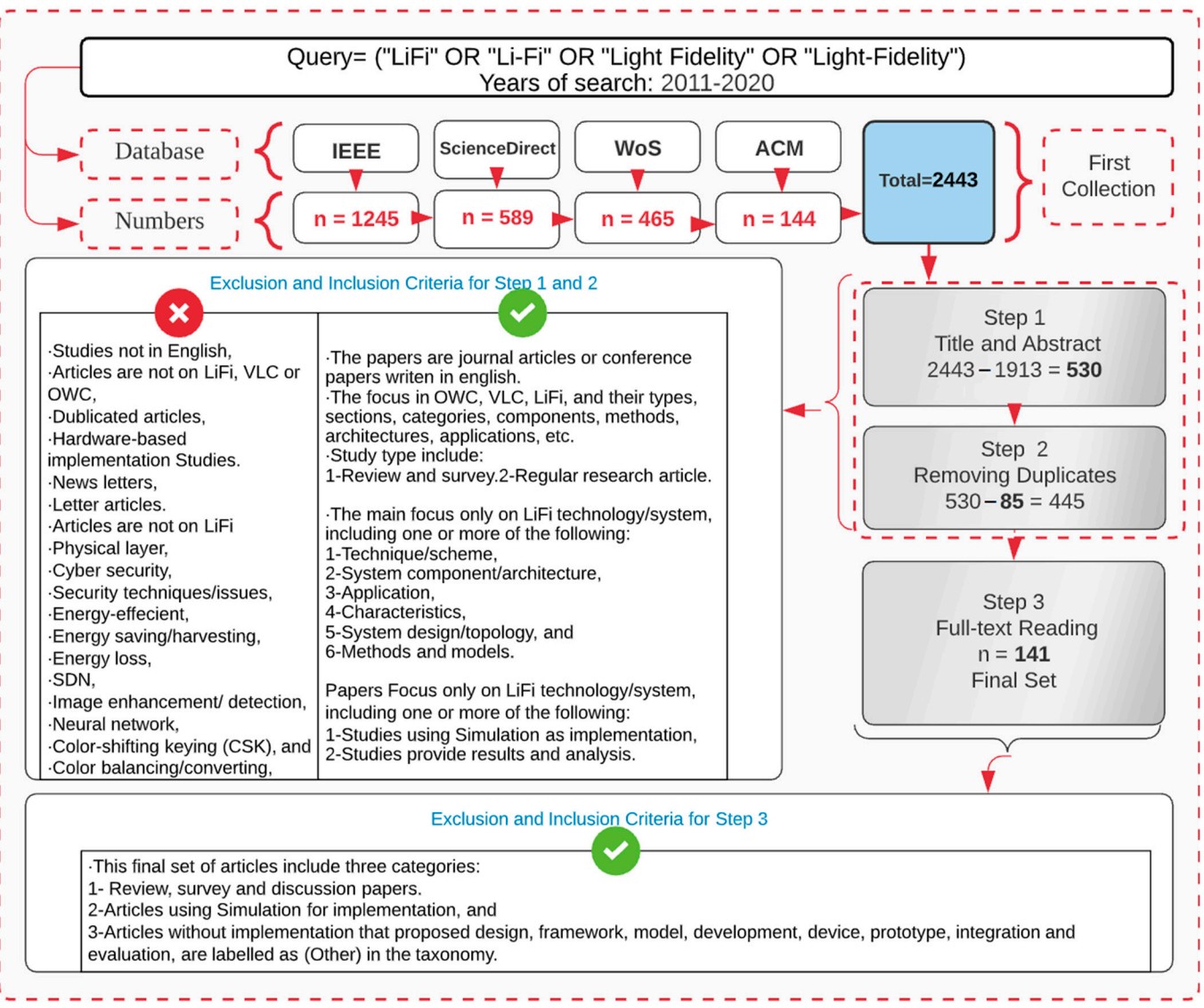

**Figure 1.** Development Study selection, including search query, inclusion criteria, and exclusion criteria.

### 2.1. Selection Method

This review relied on certain elements and settings to gather its related articles from the search stages to the final stage of the preparation of this paper. The first sub-element applied is the key words. Identifying areas of research through the use of suitable keywords is an important aspect for any study. For existing studies on LiFi technology, the most common keywords used for such a process includes the word LiFi. This keyword has been used in all research related to LiFi, including the work on LiFi by its founder Haas and other related works [14]. Thus, this keyword is deemed sufficient and suitable to cover the areas of importance in this review. All articles selected in this study were written in English. They are either journal research articles that present primary research or review studies that are also valuable sources of knowledge that present secondary studies. All papers included in this review were published in the span of 10 years since 2011, when LiFi was first introduced to the world. The applications and domains of LiFi are very diverse and wide. Hence, including them, all in one paper will be nearly impossible. Instead, selecting certain areas of research on LiFi and presenting them significantly will still yield remarkable findings. Therefore, the last element in this section will be directed to the eligibility criteria (i.e., inclusion and exclusion), which have been applied in this review. All studies included were LiFi-related simulation studies. The rationale behind our selection

is to present studies that researched LiFi using simulation. LiFi, as a technology, is still not much widely used around the world, and only certain countries and industries have adopted it in their communications. Therefore, presenting all related simulation studies will encourage researchers and developers to have hands-on testing and to experiment LiFi even before it is officially used in their countries. For the exclusion criteria, any OWC articles that do not discuss LiFi-based systems as part of their communication and hardware-based studies have been excluded.

### 2.2. Data Collection

Data were collected and extracted to analyze every article for various attributes subsequently listed and grouped in matching categories using an Excel spreadsheet. For this review, four scientific digital databases were utilized for the related article search, download, filtration, extraction, and drafting of this review. Specifically, Web of Science (WoS), IEEE Xplore, Science Direct, and ACM, respectively. These databases were deemed sufficient and appropriate to cover all LiFi technology-related studies.

### 2.3. Article Results and Statistical Information

The search and filtration processes of articles began with the initial search on four databases, using the keywords. All the excluded articles did not match the eligibility criteria of this review, which were mentioned earlier. Figure 2 represents the final set numbers of articles depending on the year of publication started ten years ago. Figure 3. Describe the types of each article, whether it's a conference or journal article. The final number of included papers that discussed all LiFi-related aspects while applying all the eligibility criteria was 141. The search and filtration processes of articles began with the initial search on four databases, using the keywords. All the excluded articles did not match the eligibility criteria of this review, which have been mentioned earlier. Figure 2 represents the final set of articles which has been based on the year of publication starting ten years ago. Figure 3 describes the type of each article, which is either a conference or journal article. The final number of included papers that has discussed all LiFi-related aspects while applying all the eligibility criteria is 141.

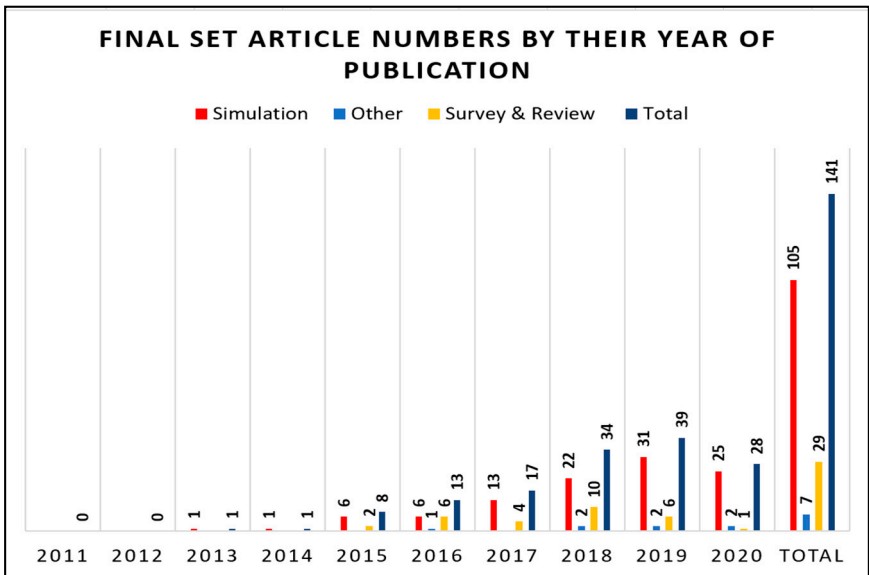

**Figure 2.** Number of articles' types based on the year of publication.

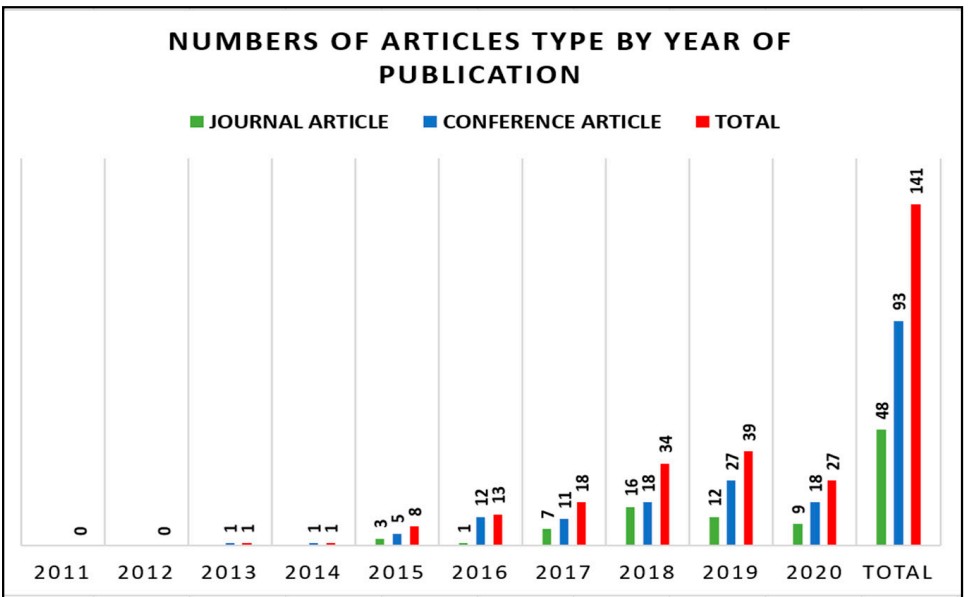

**Figure 3.** Types of articles based on their year of publication.

## 3. Taxonomy

This section aims to present the taxonomy of related articles that are selected from This section presents the taxonomy of related articles that have been selected from previous literature. All articles have been classified into two main categories and a few subcategories. The main categories discuss different types of publications, including journal articles and review studies. In the first main category, all related articles either discusses LiFi simulation or other topics. In the first sub-category, simulation-related subcategories, including (1) method, (2) problems, (3) environment, and (4) transmission direction, were discussed. Various aspects, including design, development, model, concept, and evaluation, were addressed for other studies. These categories were classified based on their relevance to LiFi simulation studies, which are the focus of the studies included in this review. In addition, creating the taxonomy enabled the categorizations of literature based on a common theme inspired by the reference and nature of the studies and agreed upon during the authors' discussions. The taxonomy analyses that contain the most important findings is presented in Figure 4. with the respective discussions.

### 3.1. Journal Article

As shown in the taxonomy, 113 out of the 141 articles were separated into two significant sub-classes based on their implementation and non-implementation which relates to: (i) Articles that used simulation (105/113) and (ii) others (non-Simulation) (7/113). This section focuses on the literature itself rather than methodologies and concepts.

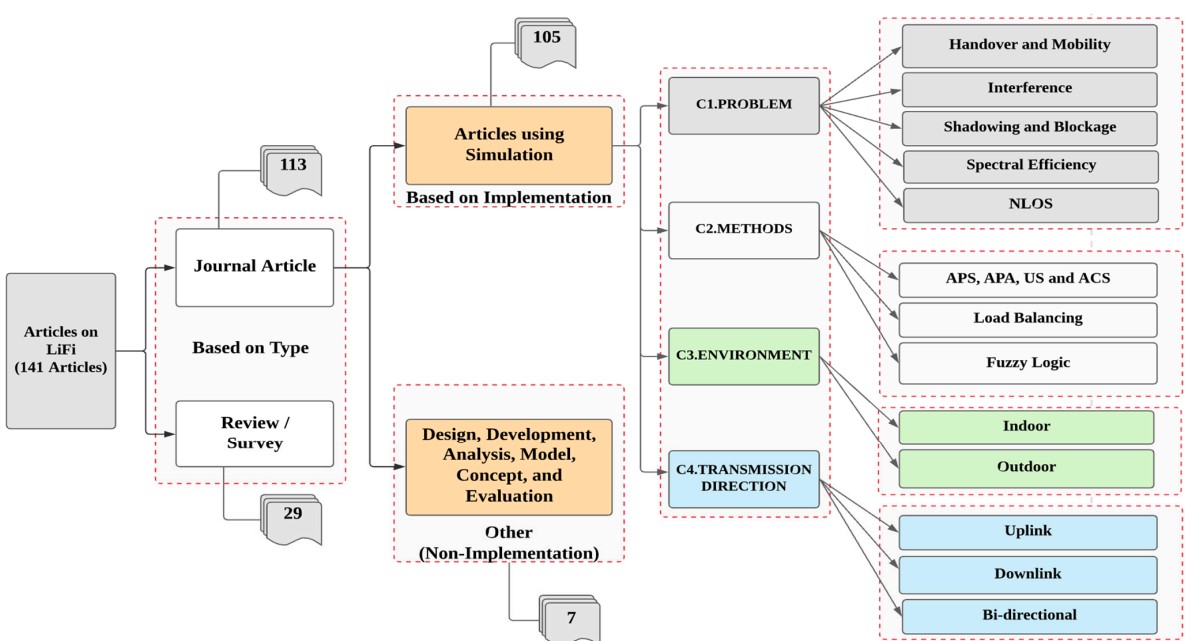

**Figure 4.** Taxonomy of literature on LiFi.

### 3.2. Simulation

The articles in this class (n = 105) were used to produce a comprehensive overview of studies on LiFi that use simulation in implementing their approach. Network simulation is a technique whereby a software program models the behavior of a network by calculating the interaction among different network entities (e.g., routers, switches, nodes, APs, and links). This section comprises four subcategories: (i) Category 1 (C1) consists of common problems in LiFi, which contains 94 articles and is mostly focused on considering issues, such as Handover (HO), mobility, and interference problems. (ii) Category 2 (C2) consists of common methods used in LiFi simulation studies, and it contains 45 articles. Most of these works used Load Balancing (LB), AP selection/assignment, and user/Atto-cell selections (ACS). (iii) Category 3 (C3) consists of common environments in which LiFi systems could be implemented, containing (n = 71) articles. The majority focused on conducting their study in indoor scenarios while the rest considered outdoor scenarios. (iv) Category 4 (C4) consists of the transmission direction of data, and it consists of (n = 48) articles. Most of the aforementioned categories focus on the downlink (DL) and bidirectional (BID) transmission, and only a few have considered the uplink (UL) transmission. In this section, the simulation studies are classified based on these four categories, as illustrated in Table 1. Many studies have proposed novel methods and schemes for access point assignment (APA) and access point selection (APS) within different environments by assessing the literature. Few of these studies considered working on UL, whereas most studies focused on the DL side of the transmission. This section focuses on simulation studies. A total of 32 out of the 105 articles involved simulation studies that used the Monte-Carlo simulation method [28,31–60]. For other studies, different simulation software, including MATLAB and OptSim [61], or MATLAB only [62,63], MATLAB and OPTI-WAVE [64], NS2 [65], TracePro [55], and Mininet emulation environment [66], were used. These four classes have been used to summarize all the previous literature on LiFi studies as shown in Table 1. All the classes and categories in Table 1 include 93 out of 105 articles which are oriented on simulation studies. Aside from the simulation studies that fall under the previously discussed problems (i.e., C1, C2, C3, and C4), other simulation studies that do not fall in the previously discussed categories still exist. To begin with, [63] proposed a novel modulation technique called Hybrid DC-Biased Asymmetrically Clipped Pulse-Amplitude-Modulated (HDAP-OFDM) power-efficient communication, as well as a comprehensive dimming control. In [67], researchers presented a network congestion characterization

related to LiFi and a High throughput satellite (HTS) under the Ka-band modulation schemes during adverse weather conditions in tropical regions. The study in [68] proposed a fascinating system for massive data transmission that could be performed by free-space communication and was proven to be helpful for links within high and large buildings. The respective data transmission capability can be reached via the big data concept [69] and developed a smart vehicular communication system that protected vehicular collisions on the roads using LiFi technology. The study in [15] simulated a Direct Current Optical Orthogonal Frequency Division Multiplexing (DCO-OFDM) modulation-based MIMO system used for LiFi technology. In [70], the authors designed a smart traffic system to communicate between automobiles and their traffic environment. This system is usually called V2X technology. In the work of [37], they proposed a new modulation format called HDC-OFDM for LiFi by combining the aspects of the existing Diversity-combined asymmetrically clipped optical OFDM DACO-OFDM and DCO-OFDM. The proposed HDC-OFDM has the benefits of the power efficiency of DACO-OFDM and the dimming flexibility of DCO-OFDM. In [50], the authors explored the potential of transmitting signals over in-phase and quadrature IQ components separately using two LEDs and Wavelength Division Multiplexing (WDM), namely, IQ-WDM. By sending baseband signals from the IQ components, the higher attenuation due to the higher frequency band can be avoided. In [51], the authors channels with reference to the proposals for the standard mandatory PHY of the IEEE 802.11bb. In [55] a simplified analysis was done. However, the general case was also presented where ordinary commercial LED bulbs were analyzed in terms of their light output as compared to the receiver threshold. In [71], they also presented a simplified yet general case in which ordinary commercial LED bulbs were analyzed in terms of their light output as compared to the receiver threshold. In [72] the evaluation of three relevant scenarios with RF/LiFi hybrid networks was done to determine the improvement possibilities provided by this technology and was assessed for the potential of the respective realistic developments.

### 3.3. Other Articles

This section presents other articles (n = 7) that did not conduct any implementation but proposed various useful aspects and analysis in the field of LiFi technology. These aspects are outlined as follows: (i) Design: The research done by [129], proposed a new architecture to leverage VLC and a steerable IR beam to enhance the data transmission rate of indoor users in the Fi-Wi access networks. The study in [130] proposed a novel idea using balloons in LiFi based disaster management or during a natural calamity. This proposal has been taken as a use case to provide a safe and cost-effective solution for these disaster management cases. (ii) Development: The study in [131] presented an automated system that detects drivers in a drunk condition to provide an effective solution and an intelligent system for vehicles that can prevent accidents due to this cause. Meanwhile, the study in [132] proposed a flexible system concept for LiFi that addressed the heterogeneous demands of future IoT applications while using harmonized technology. Their objective was to present their system by introducing a standard set of core technologies and combining them in different ways to build specific solutions. (iii) Analysis: The study in [4] aimed to retrieve the most promising applications, such as those used for homes, supermarkets, museums, and communication among vehicles, through the analytic hierarchy process model and qualitative analysis. (iv) Model: [133], offered five PAMS information system models that ensured data collection and transferto the crisis decision-making centre. The purpose of these models was to build and study the model options of an information system designed to transfer data relating to an accident, process the data received in the crisis centre, and make operational decisions to ensure enhanced safety and eliminate emergencies. (v) Concept: [29] described LiFi and highlighted its key differences to VLC. It also discussed misconceptions and illustrated the potential impact of this technology on several existing and emerging industries.

**Table 1.** Classification statistics of simulation studies based on multiple aspects.

| | C1 Problem | | | | | C2 Method | | | C3 Environment | | C4 Transmission Direction | | |
|---|---|---|---|---|---|---|---|---|---|---|---|---|---|
| | HO and Mobility | Interference | Shadowing and Blockage | Spectral Efficiency | NLOS | APS, APA, US, ACS | LB | Fuzzy Logic | Indoor | Outdoor | UL | DL | Bi-Directional |
| Ref. | [15,17,19–21,23,30, 37,41,44, 50–55,57,61–66,68–70,73–80] | [19,33,35,42, 48,54,57,60–62,66,74,79–102] | [31,32,43,53, 56,57,73,81–83,103–109] | [74,87,96, 98,103] | [39, 105] | [11,15,34, 40,50,51, 53,56,61, 62,64,66, 74,75,78, 80,91,100, 108] | [2,17,21,26, 28,40,50, 56,62,64, 66,69,70, 74,75,78–80,87,88, 91,95] | [38,53,56, 110] | [2,9,11,15,17, 18,20–23,25,26,28,30, 37,40,41,50,51, 53–56,59,62–66,68–70,73–76,78–80,83,84,87–95,97,100,107, 108,111–123] | [75,87] | [49,95, 124] | [2,9,32–35,39,41,47, 48,56,62,66, 74,76,77,79–81,84,86,89, 97,98,109, 112,113,115, 116,121,125–128] | [16,36,52,61, 78,87,90,102, 103,105,111] |
| Total | 34 | 36 | 17 | 5 | 2 | 19 | 22 | 4 | 67 | 2 | 3 | 34 | 11 |
| Total | 94/105 | | | | | 45/105 | | | 69/105 | | 48/105 | | |

### 3.4. Review and Survey Articles

The articles in this class (n = 29) discuss the efforts of previous researchers in doing extensive reviews or surveys. All these review and survey studies are discussed in Table 2 with respect to their year of publication, their main contributions, and from which database they have been extracted. To the best of our knowledge, many reviews and studies in the LiFi area have been published. Most of these works highlighted many important aspects that are related to the new technology. In addition, these works focused on many common important points in this domain, including challenges, limitations, and issues faced by new area. Important differences between LiFi and current RF technology were also highlighted. Meanwhile, other studies focused on the difference between LiFi and OWC. Subsequently, several of these works found the need to discuss the new system's architecture to provide an overview and detailed information to other researchers in this area. A possible reason for such an aim was to come up with more solutions to solve LiFi-related problems.

**Table 2.** Literature review of review and survey articles.

| Main Contribution | Ref. | Source |
|---|---|---|
| The applications of VLC, including LiFi, were analyzed. The disadvantages and advantages of the VLC method across LiFi, Wi-Fi, and Wi-Max wireless networking were contrasted. | [134] | IEEE |
| Standardization groups that work in VLC technologies and state-of-the-art research on LiFi networks were discussed. | [10] | WoS |
| The general functionality of WiFi and VLC or LiFi was described, and a functional structure for the coexistence of both technologies was illustrated. The ongoing research effort was addressed, and current and future research challenges were identified. | [135] | IEEE |
| The authors highlighted the data transmitted by LEDs, previous studies and research technologies established, modulation methods, deployment, and challenges in LIFI applications. | [136] | IEEE |
| The authors introduced a description of LiFi in terms of working principles and LiFi features, such as capacity efficiency, safety, security, applications, and advantages. | [20] | IEEE |
| LiFi features, such as working principles, challenges, and applications, were presented to compare Wi-Fi technology. | [26] | IEEE |
| The importance of LiFi in construction, bulb component functions, applications, and its disadvantages and advantages were highlighted. | [137] | IEEE |
| LiFi technology was presented in terms of work principles. The efficiency, similarity, and information transmission of LiFi through LEDs were compared with those of WLAN. | [138] | IEEE |
| The authors highlighted the VLC potentials, the current RF problems, and LiFi capabilities on improving indoor data transfer and connectivity. In addition, LiFi concepts, implementations, weaknesses, and difficulties were addressed. | [139] | IEEE |
| This paper presented the LiFi technology development, including design, modulation, efficiency and challenges, difference LiFi and WiFi, LiFi topology finally, LiFi physical and MAC layer. | [140] | WoS |
| The LiFi state-of-the-art development, strengths, and weaknesses were emphasized. Moreover, LiFi challenges that were faced in the implementation of communication techniques and the modulation used in LiFi were discussed. | [27] | IEEE |
| Authors motivated the need to learn and understand new technologies (e.g., LiFi) and their advantages. Working smart meters were introduced, and communication information was recorded daily to the utility for monitoring and billing purposes. | [24] | IEEE |
| The authors reviewed a dynamic LB technique for hybrid LiFi/RF networks, and three types of LB algorithms that were proposed in the literature were compared. Moreover, they discussed open challenges in this area of hybrid LiFi. | [141] | ACM |
| Offers an extensive analysis of recent articles and advances in LiFi. LiFi-based IoT architecture was also introduced. | [6] | IEEE |
| LiFi applications and experimental devices for LiFi communication, as well as the critical paths for the current innovation, were presented. | [3] | IEEE |

**Table 2.** *Cont*.

| Main Contribution | Ref. | Source |
|---|---|---|
| Recent research on LiFi was summarized. The comparison between LiFi and Wi-Fi was introduced. The weaknesses, strengths, and application of LiFi in a different environment were addressed. | [5] | IEEE |
| This work focused on developing LiFi technology. LiFi applications, advantages, limitations, and comparison between LiFi and Wi-Fi were discussed. | [7] | IEEE |
| Issues regarding existing LiFi systems and monitoring applications were discussed. Moreover, different applications were introduced for various places, such as rooms, homes, or supermarkets. | [142] | IEEE |
| LiFi with a 5G system was addressed. The benefits of the network layers of the latest technology, which also features LiFi, business, and industry, such as industry 4.0 and myths regarding LiFi, were addressed. | [14] | SD |
| The hybridization of LiFi and Wi-Fi Li+ WiFi was discussed. Correlations, when Li + WiFi come together, were highlighted. Upcoming researchers were given directions to link these technologies. | [143] | IEEE |
| State-of-the-art was investigated, and effective physical layer modulation schemes were explored in the context of indoor positioning. The main modulation techniques for LiFi were investigated. | [144] | WoS |
| An overview of OWC technology, such as VLC, LiFi, Optical camera communication OCC, free-space visual communication, and light detection and ranging, was provided. The primary goal of this study is to highlight the differences among wireless optical technologies. | [21] | IEEE |
| The authors reviewed state-of-the-art LiFi technology and discussed the system throughput and the user's QoS in heterogeneous networks. | [22] | WoS |
| The necessary foundations of LiFi were discussed, and the possibilities of incorporating LiFi in a communication system were investigated by defining the architecture. Moreover, they discussed transmitter-to-receiver data transmission. Furthermore, LiFi and other technologies, such as Bluetooth and WIFI, were compared. | [145] | SD |
| This survey introduced a navigation system that supports blind people by sending audio feedback. In addition, future scope and many studies on the navigation system have been discussed. | [146] | IEEE |
| This survey introduced LiFi communication techniques and numerous previous architectures to utilize VL. The main difficulties faced in achieving optimum VL contact, and the performance analysis summary of various LiFi architectures, were presented. | [147] | IEEE |
| This paper supported the need for a new spectrum of emerging and potential wireless networks. The authors introduced a taxonomy that defined the four significant developments in this field (e.g., VLC, OCC, Free-space optical communication FSO, and LiFi. | [148] | IEEE |
| The improvement of LiFi compared with existing devices was explored. Moreover, misconceptions of LiFi network modeling were presented through dissection. | [149] | SD |

Furthermore, many studies emphasized the advantages and disadvantages of LiFi and highlighted its applications. Our work found that no study focused on all these important points together in one article, and no study has discussed the classification of the simulation studies. Considering that no study mentioned the country of publication and the publication trend in this area, therefore in can be deduced that no study has reviewed all the surveys and articles in this area. Thus, our study aims to address this and introduce an overview of all the important aspects of this area in a consolidated form.

## 4. Applications of the LiFi Network

This section discusses a detailed overview of LiFi applications in various areas from the articles included in this review. The discussions are categorized based on their mass usage in stipulated areas.

### 4.1. Transportation

LiFi can be integrated into many areas. One that has gained momentum is transportation. Research application topics in this context included Vehicle-To- Vehicle communication (V2V) [53], Vehicle-To-Person communication (V2P) [4], Vehicle-To-Infrastructure (V2I) [150], and Vehicle-To-Network communication (V2N) [150]. Based on the nature of these applications, the role of LiFi technology from the literature has shown to be among the most critical ones, mainly because of the capabilities of LiFi to function as a real-time communication technology. Information can also be shared through car headlights. Therefore, contact can be generated between the driver and the vehicle. Furthermore, LiFi has massive potential in other ranges of car applications because it enables extensive collaboration and crash avoidance [17]. In traffic signals, LiFi can also be attached to vehicles via LED lights, thereby helping in handling traffic in geographical areas which has high density of cars more effectively. This technology can assist in the continuous regulation of traffic and reduce pollution and injury frequency. For example, LED car lights with LiFi sensors can warn drivers when other cars come too close. Radio waves interfere with PCs and cell phones. The interfering signals could also be unsafe and might affect the safety of a patient [17].

Furthermore, Wi-Fi is unauthorized in certain domains because of the interaction among signals such as those in radiation operating rooms. Wireless Internet in several hospitals can obstruct the signals of control devices. LiFi solves these issues, often in operating rooms when the network is constant rather than flickering. LiFi is ideally used in modern medical facilities. Thus, addressing the challenges, including interference, which occur when using wireless technologies, such as IR and RF. Thus, LiFi is a substantial complementary approach to radio, being particularly attractive in scenarios where RF and/or IR fail to provide security, privacy, and zero electromagnetic interference [151]. This avoidance of electromagnetic radiation challenges which occurs with other wireless technologies in locations like aircraft, surgical facilities, and the oil and petrochemical industries enables many new opportunities [22,152].

### 4.2. Sound System Communication

Other types of LiFi applications include sound system communications. Studies on sound system applications are fewer as compared to other applications. However, they are discussed because of their significance in integrating LiFi. In [150], various coloured LEDs may relay sound and even execute an audio signal transmitting scheme at the Pulse width modulation (PWM) base [153].

### 4.3. Location Detection

Location detection, which is used to process data position to a very comprehensive and precise degree is another application of LiFi. LiFi technology offers indoor (10 cm high-point) navigation information and guidance, which is useful for recognizing the users' faces [42]. Therefore, LiFi may be used to locate the user's position [82,154]. For example, positioning data are obtained by LiFi because the user is directed through the details of the position. At that point, the AP sends the device position information to the central server. Subsequently, the Central Server utilizes the user position details to create a user location map (location information) [155].

### 4.4. Educational System

LiFi has performed in similar outstanding standards as other technologies used in a broad range of applications, such as education, eventhough LiFi is among the latest of technology. The most advanced Internet access technology offers one of the fastest Internet connectivity [70,143]. Thus, LiFi can be considered the most modern Internet networking infrastructure that provides the best connection to the Internet. It can be an excellent alternative to Wi-Fi in educational and corporate organizations. Therefore, it

can be considered among the ideal frameworks to be used by colleges, lecture rooms, conference rooms, testing centres, research centres, and laboratories [156,157].

### 4.5. Indoor and Outdoor Lighting

Any light source, such as streetlights, may deliver LiFi hotspots. Indoor and outdoor lights in the office, home, health facilities, business sites, airlines, cars, and streets allow artifacts to be linked to the Internet using attractive VLC features. Therefore, street signage can be used as a hotspot and can be rendered as a proper LiFi application. Moreover, the same networking and sensor technologies may be used for lighting and data management [17,82,150,154].

### 4.6. Industrial Information's Environment

In industrial areas, data needs to be transmitted properly due to the intensity and complexity of the interconnections of devices and applications. In this context, LiFi can exchange slipper rings, sliding connections, and short cables, such as commercial ethernet. It is also an alternative to furnishing the increasing industrial wireless requirements because of its respective real-time capabilities [23]. Wireless data are transmitted by super-fast modulating light and the process is invisible to the human eye. In addition, the system transfers large volumes of data at high speed offering benefits, such as high protection and reliability, without losing wireless connection This is mainly because of its ability to automate the production process through its excellent connectivity and remarkable data transfer. These features enables LiFi to have considerable effects on business. Industry convergence helps suppliers to handle changes in technological developments. Nevertheless, questions such as how do the performance of systems get affected by technologies such as LiFi and how does LiFi plays an important role in other business roles must be addressed. LiFi, which is recognized as a key player in the Industry 4.0 ecosystem, can affect the signnificance achivhed by the next industrial revolution [119].

## 5. Component and Architecture of a LiFi System

LiFi, as a potential and strong candidate in communication, is different from traditional Wi-Fi in many aspects, such as its components and architecture. Understanding such aspects will aid in comprehending and utilizing the technology efficiently in the future. Therefore, this section aims to elaborate and discuss the components and architecture of LiFi. A standard LiFi operating system involves propagation in one path, such as DL [89,158,159] or UL [34,155] or where both of them are BID [78,103,160]. Power and data may be supplied to each light fixture through various approaches, including Power Over Ethernet (PoE) and Power Line Connectivity (PLC) [14]. An UL is introduced by utilizing a transmitter on the user equipment (UE) or sometimes using an IR source and a receiver near the light fixture. These light fixtures, which simultaneously serve as wireless LiFi APs, generate an incredibly tiny cell called attocell. This tiny cell has a high bandwidth density because of a single light source [161]. The key and typical components for data transmission and reception in DL and UL are specified in this section. The transmitter and receiver also run simultaneously between the UE and the AP. Similar techniques, processes, device architecture, and configurations were suggested by most researchers who worked on improving LiFi systems. Consequently, the most common components used in LiFi systems in DL and UL procedures are introduced in this section.

### 5.1. Downlink Transmitter DLT

A LiFi light bulb is used as a DLT [162]. Usually, the light bulb consists of one LED or many shaped clusters [42,90]. Quick and massive current variations can be created to produce the optical output because only Light is used [163]. The LiFi light bulb is constantly operating and interruption-free because the LED can be turned on and off easily, and the number is quickly modulated. Thus, even though it flickers, the human eye is not able to see it [1]. This transmitter, known as the AP, can be connected to the receiver in UEs,

such as PCs, laptops, and mobile phones [94]. Similar to all other wireless networking technologies, this AP spans a small region and transforms IP network knowledge into bits. Also, the coverage area created by LiFi AP is referred to as the attocell [66,79]. Figure 5 illustrates the LiFi AP attocell. A few important factors must be considered in DLT. (i) The line of sight (LOS) component: the contributions of the received signal, wall, and surface which are reflected in the light. Therefore, the LiFi technology is not strictly oriented to the LOS [17,120,164]. However, LOS connections offer greater results than the Non-Line Of Sight (NLOS) link. (ii) Presenter of light: This factor depends on the irradiance angle and the environment. Most LiFi simulation studies assume that the irradiance angle in their device design is perpendicular to the floor [49,66]. This phenomenon shows that the attocell for each AP is in an asymmetric circular form. Meanwhile, the irradiance angle is a significant feature for designing and creating methods, algorithms, and schemes, such as APS, LB, and HO management and coordination. (iii) Illumination: This factor refers to the important volumes of transmitted optical energy. This power follows the transfer of electric power to optical power [66]. Moreover, the room's height has a major effect on the optical power, which contributes to the intensity of the AP–UE communication relationship. However, more APs require more lighting, subsequently increasing the noise between the APs. Thus, the Signal-To-Interference-Plus-Noise Ratio (SINR) must be considered. Although all these factors might seem identical, they are not.

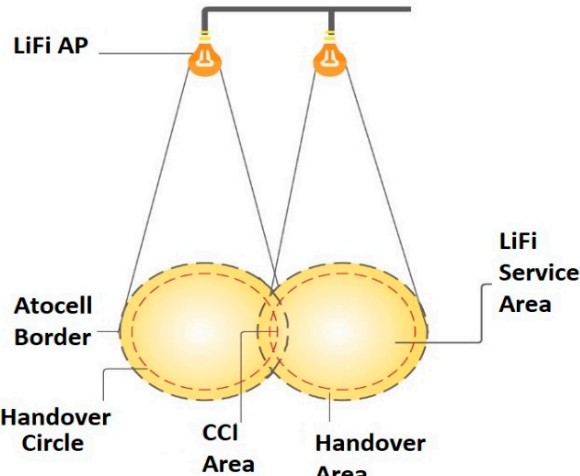

**Figure 5.** LiFi attocell in DLT.

However, they participate in AP assignment and communication resource allocations. The LiFi AP covers a specific and limited area. Moreover, attocells are cells greater in scale than the communication in the LiFi network. A strong spatial spectrum quality can be obtained by reusing the bandwidth in each attocell [94]. The overlap zone, Optical Co-Channel Interference (CCI), is recognized and treated as noise because the same bandwidth is reused by all attocells [66]. The attocell diverges scale based on the source of illumination and the form of light. (iv) The distance between the AP and the region of irradiation. The attocell boundary represents the edge of the illuminated region. CCI occurs when two or more attocell users are linked as long as they remain inside the HO circle. This phenomenon is the same as that in the LiFi service area. Thus, users will connect to the AP and receive a LiFi signal from there. Other signal forms from another AP, such as Wi-Fi, may affect the area size [66]. As users across the HO zone borders, they would experience HO where they would remain inside the HO field. Depending on the algorithm used, such as the LB algorithm, this approach requires multiple milliseconds to seconds [66,79]. However, even within the HO zone, users will not experience HO in certain cases. This instance, which is termed as HO overhead [40,62,66,165,166], may occur even beyond the HO area because of other factors, such as: (i) the capacity of networks, (ii) the density for devices or users, (iii) signal intensity strength, and (iv) availability.

## 5.2. Downlink Receiver DLR

LiFi technology typically utilizes LEDs and Photodetectors (PDs) as transmitters and receivers, respectively. However, camera PDs or Image sensors (ISs) are used as physical receivers. IR, VL, or Ultraviolet (UV) spectra are used as a communication media [21]. Compared with the conventional usage of electromagnetic waves, such as Wi-Fi, LiFi utilizes light as a medium. LiFi signals cannot penetrate walls, and it is usually carried out using white LED bulbs on the DLT [167]. Different components, such as I PIN PDs, may be used as receivers [168,169]. An intrinsic zone is inserted between two vast regions.

When the volume increases, an incident photon can be generated by: (i) electron-hole pairings and (ii) silicon APDs [170]. A high reverse bias is added to raise the electric field and the velocity of the carriers responsible for the avalanche effect. Owing to the phenomenon of avalanches, APD shows great photo sensibility relative to a PIN PD, (iii) SPADs, (iv) CMOS sensors [159,171–173] and (v) solar photovoltaic PV cells [156,161,174]. Several benefits and drawbacks of using a PV-based receiver are shown in Table 3. Figure 6 shows the DL transmitter and receiver in a room environment. For example, from LiFi AP to cellphone: (d) refers to the distance between the AP and UE, (z) refers to half the angle of the radiation angle, and (a) refers to the angle of the receiver.

**Table 3.** Benefits and drawbacks of PD- and PV-based receivers.

| | Benefits | Drawbacks |
|---|---|---|
| **Pd-Based Receivers** | 1. Elevated bandwidth.<br>2. The output voltage of the illumination is linearly related to the transmitted optical LED power. PD can detect low-level light variations.<br>3. The avalanche effect increases the identification of slight changes in light intensity.<br>4. Low noise and high bandwidth are produced. | 1. Poor surface detection.<br>2. Requires the use of a very high-performance, high-cost circuit amplifier.<br>3. The power received at the PD is affected by the channel gain and noise. |
| **Pv-Based Receivers** | 1. LiFi receiver with auto power (neutral energy).<br>2. Allows energy harvesting detection of LiFi signs.<br>3. Broad surface detection.<br>4. Allows partial shading to be avoided and angular acceptance to be larger than that of PD. | 1. Low bandwidth standard.<br>2. A small spectrum of light variations is earned (i.e., the received light interpretations for preserving a close linear relationship with the difference in output voltage are significantly low). |

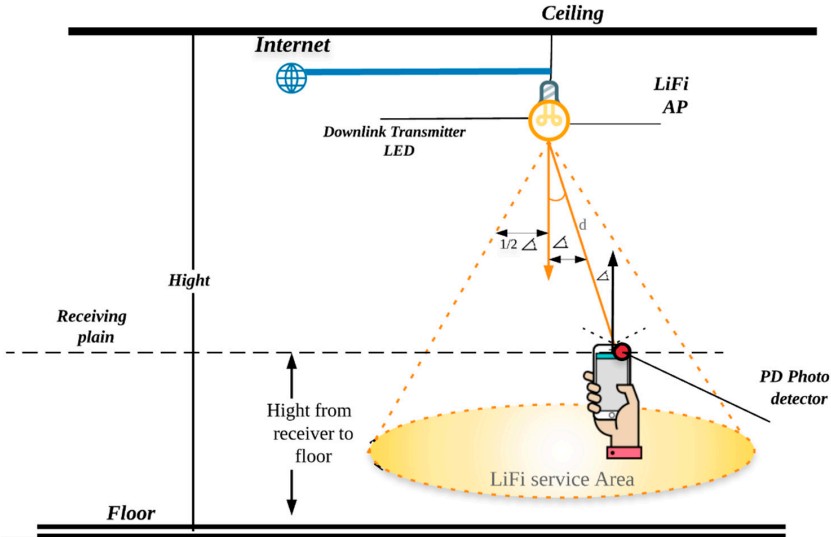

**Figure 6.** DLT and DLR in a room scenario.

*5.3. Uplink Transmitter ULT*

The efforts to achieve the UL counterpart in LiFi links and Internet connectivity for futuristic, intelligent applications also play a crucial role in designing LiFi communication systems. Despite the great work dedicated in these areas, few studies have introduced UL solutions [39,102,111,124]. Most studies on LiFi used IR and Wi-Fi for ULT [39]. IR sources with a wavelength of more than 1500 nm are safe for the human eye, which is an additional feature that supports future communications [110]. IR can be used in the UL so that the illumination constraints of a room remain unaffected, and interference with the VL in the DL will be avoided [76]. Similar to VL, the IR emission is susceptible to being blocked by room walls, thereby offering security against eavesdropping. The UL will be established with the IR source built into the user's device. The visible spectrum is not used as the UL because of the alignment difficulty of the latter and the glare from the user's device, which causes visual inconvenience. Different wavelengths are used in the DL and UL of LiFi networks, VL, and IR. Thus, the UE PD is tuned to VL and cannot sense the channel when another UE transmits via IR [36].

*5.4. Uplink Receiver ULR*

Using an IR light for UL can be a potential scheme for LiFi systems because IR light is invisible to the human eye. Using RF as a UL increases the complexity of combining space and time to increase the number of users who can access the network with a target value of rate per user. Managing transmissions by multiple users are necessary. To assure connectivity, UL coverage, and sufficient rate, multiple APs must be considered because the sole use of an AP may be inadequate. The opportunity of having the same signal copy at a different AP may induce interference if not adequately regulated through an access strategy. The ULR is called an IR detector, and it is typically placed in the ceiling at the AP beside the DLT, depending on the design that is being used. The CU controls both entities. In [95], the authors proposed a design for a UL with multiple users which consisted of five photodiodes. The UE refers to the user's pieces of equipment, such as laptops, phones, or any other devices. A cell phone is supposed to face upwards for proper ULT because of the Vertical handover (VHO) and a complicated power allocation are required. The ULT greatly depends on the user and AP locations. For instance, when WiFi is used for UL, users can move more freely in the room without considering the device orientation. By contrast, UE should be placed upward to maintain a stable connection for systems that use IR for UL. An IR ULT smart Time Division Multiple Access (TDMA)-based scheme was proposed in [95]. The authors proposed a scheme with hybrid access that aimed to ULT and/or DLT are essential for a complete LiFi system. In Figure 7, all the hardware components used in building, designing, or modelling a LiFi system network are proposed. To build, design, or study the architectural components of a LiFi system, understanding and investigating all the entities that must be included in the development of the LiFi systems are important. Figure 7 illustrates the essential components of infrastructure, AP side and UE side.The AP side includes Internet access [6], lamp driver, and light bulb, which represents the AP [137]; CU to control the flow of the data and support the LB and APA process [94]; and PLC or POE, which make to the use of the power lines as the medium of communication in the infrastructure among Internet servers [16,18,143,174], CU, and APs possible. The UE includes cell phone, PC, or a dongle, including a light detector, such as a PD or camera [118], solar cell [175], or PV [174,176], as receivers.

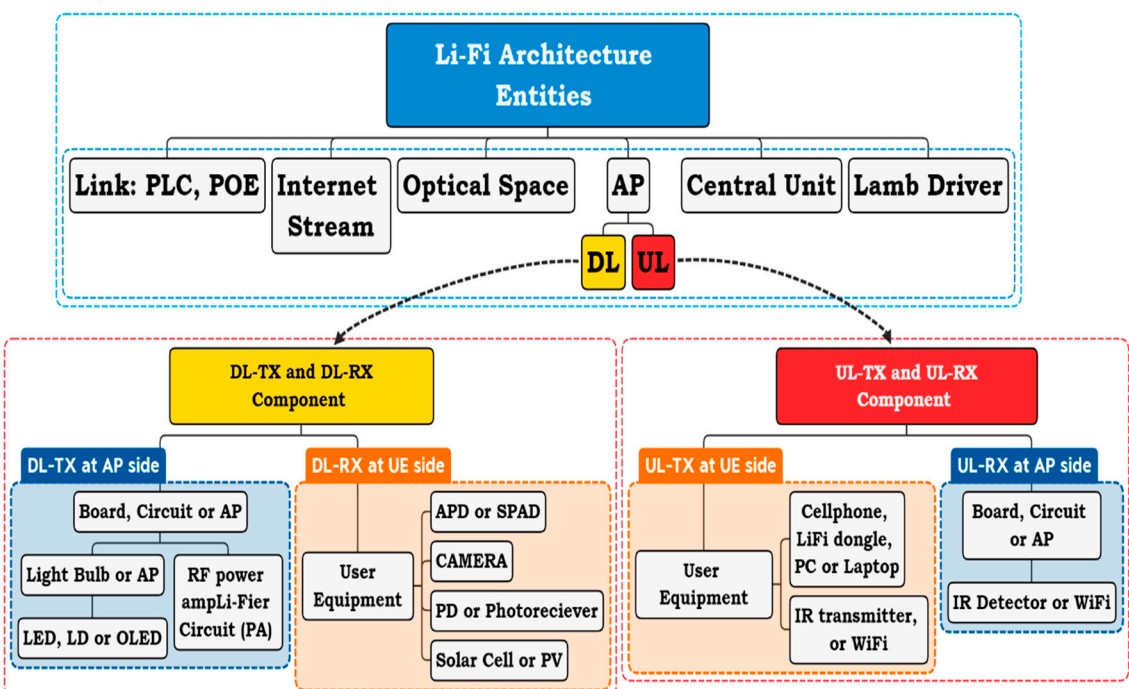

**Figure 7.** LiFi system components.

## 6. Advantages and Disadvantages

This section discusses the main disadvantages and advantages of LiFi. LiFi has low power consumption because the ULT is built to have a low peak to average transmitter ratio, as well as a lower computational complexity [34]. Furthermore, it has a fast speed link in comparison with other technologies and a high data rate. Thus, LiFi could reach a very high degree of spectral efficiency because of the massively reduced interference by the co-channel in dense network implementation [177].

LiFi is an accessible system because it uses the existing infrastructure, including lamps and lighting components that are deployed in almost all indoor areas. It is also considered a green technology because light has no harmful effects on the environment [14,110,139,144]. This feature is non present in the RF-sensitive applications that may have potentially negative health consequences for humans, such as high frequency, come from 5G networks [152]. LiFi connections also serve as an enhanced and reliable link over the infrastructure (e.g., 5G) based on a V2V framework. LiFi can alleviate some important challenges faced by 5G and is ready for seamless integration into the 5G core [5,143,145,178].

The electromagnetic spectrum VL size is greater than 300 THz. Similarly, in these networks, a broad, unregulated bandwidth would ideally be used, and this bandwidth is 600,000 times faster than a 500 MHz WiGig channel, which reaches up to 7 Gbps (wireless gigabit alliance). Moreover, research shows that the achievable data rate of a multiple LED device when utilizing WDM will exceed 100 Gb/s [57]. The combination of RF and LiFi systems has a major benefit because the LiFi system offers high data speeds coupled with lighting, and the RF system provides pervasive coverage. As an environmentally sustainable and renewable networking technology with unrestricted bandwidth resources, the VLC framework has emerged as an alternative to traditional wireless technologies. The VLC device has many other special features, including a broad unlicensed bandwidth that can deal with a crowded radio spectrum. Additionally, it offers low-cost facilities because it is used as a transmitter by LEDs. Furthermore, it is free of interference from modern and existing technologies, and it has a stronger security physical layer.

Apart from all the benefits that have been stated, it offers illumination without the added cost, and, most critically, it avoids any sort of health issue [5]. There are Similar to the issue of the limited range, LiFi AP are approximately 10 m shorter. A small field of

coverage, as well as blocking [106] and shadowing [57,115], is clearly observed. There is an issue with LOS [179] and possibly noise from another neighbour with illumination or other AP [180]. In a realistic view of LiFi systems, the DC forward current of the LEDs is higher than the data-carrying signal to satisfy the criteria for illumination. Therefore, the total interference is controlled by the photo-induced shot noise [177].

LiFi has a high-security connection because light cannot penetrate the walls [21,181]. This phenomenon would provide the user with a stable link connection and a decent data rate. One of LiFi's greatest benefits is its low-cost network infrastructure because it is also easily assimilated with the existing lighting scheme due to its inexpensive equipment [21,88]. Furthermore, LiFi is not regarded as involved in high-power consumption because it is environmentally sustainable but not costly [182]. Moreover, no other electromagnetic infrastructure network interferes with the LiFi system because of the different electric wave frequencies [21,23]. Gamma rays-X and UV are not used in LiFi because they are dangerous to human health [23]. The SNR of LiFi is large, achievable, and can be integrated into the new lighting method effectively [21,88].

A further advantage of the LiFi system is that it can work underwater because light can pass through fluids [88,109]. It also offers efficient and effective communication because its bandwidth is massive. Unregulated LiFi offers high data transmission speeds for indoor applications with high-speed communication. With a frequency of over 1 MHz, the LiFi operation is very high quality [21,23,88], subsequently making the system very effective [23,183]. In reality, if anyone goes beyond the LED source, the signal is lost because of barriers. LiFi also does not work when fully obstructed (when there are no LOS nor NLOS paths) [183,184]. Given that light cannot go through objects, the signal is immediately cut off; this phenomenon is called a blocking issue [23,57,106,115,183,184]. The reliability and coverage of the network are major challenges [23] because of the reflection of light [88].

## 7. Comparison between LiFi and other OWC/RF Technologies

This section emphasises the two types of comparisons: (i) LiFi and RF, and (ii) LiFi and other OWC technologies. OWC technologies have four types: (1) FSO, (2) VLC, (3) OCC, and (4) LiFi. Typical applications for each of these technologies can be classified according to four basic parameters: (1) link data rates, (2) range, (3) duplex mode, and (4) communication mode. Study [148] introduced a taxonomy that clearly showed the four types of OWC. All four types of OWC do not interfere with RF and are safe.

The OWC spectrum is unlicensed and offers better security [148] compared with RF-based wireless systems. However, many differences among them are noted, as shown in Table 4. Table 5 shows a comparison of these technologies and explains the main differences between the LiFi system and the RF system in terms of elements, functionality, specifications, and other factors. Moreover, there are significant features and differences between LiFi and VLC; they both use existing illumination infrastructure [2,88,129]. The availability of VLC or LiFi attocells adds a new tier, creating three-tier networks [185]. LED luminaires or laser diodes (LDs) are used as transmitters, and photodetectors (PDs) are used as receivers in VLC. It can provide communication, lighting, and localization while using simply VL as a communication medium. VLC does not require mobility or illumination assistance. Sunlight and ambient light sources significantly impact VLC performance, rendering it inappropriate for outdoor applications.

LiFi, like VLC, employs LEDs or LDs as transmitters and PDs as receivers, as well as communication, illumination, and localisation. A LiFi system has a transmitter and a receiver at both ends of the transmission, allowing for bidirectional communication. Furthermore, it facilitates point-to-multipoint connections. Mobility and lighting support are required in LiFi. Sunlight and ambient light sources have a significant impact on LiFi functionality [185].

The studies [140,186], claims that LiFi and VLC both use a light-based technology for data transmission. VLC differs from LiFi because it is a unidirectional, point-to-point light

communication system with lower data speeds. While LiFi is a networked, bidirectional, and high-speed wireless communication system. In contrast to VLC applications, infrared light is being examined for communication in LiFi and communication distances in LiFi limed to 10m while in VLC is 20m; moreover, VLC uses VL spectrum as a transmission medium while LiFi uses VL, UV and IR. Furthermore, LiFi uses PD as a receiver, while VLC uses PD and Camera. Moreover, LiFi and VLC have different standardization; LiFi standardisation is IEEE 802.11 bb, while VLC standardization is IEEE 802.15.7 [21,140].

LiFi and VLC both offer high data rates, but WiFi and small cells give relatively broader coverage for increased support for mobility. If LED transmitters are situated closely, VLC and LiFi systems suffer from interference effects [187]. In hybrid systems, similar to WiFi/VLC, WiFi/LiFi, and small cell/LiFi systems, macrocell/VLC and macrocell/LiFi hybrid systems also support offloading traffic to high-data, cheaper VLC and LiFi networks to enhance spectrum use, dependability of links, seamless movement, and security for optical wireless users. VLC and LiFi VLC are dislodging the increasingly large traffic in the macrocellular network. Traffic is routed across two networks in RF/VLC and RF/LiFi hybrid systems, ensuring improved service levels and maximising the use of resources. A hybrid system network can be selected based on the parameters such as traffic type, needed safety levels, required data rate, lighting requirements, mobility assistance and uplink/downlink service type [185].

**Table 4.** LiFi and other OWC comparison [19,21,68,148,162,188–193].

| METRICS | FSO | VLC | OCC | LiFi |
|---|---|---|---|---|
| Indoor/outdoor | Outdoor | Indoor/outdoor | Indoor/outdoor | Indoor/outdoor |
| Outdoor/indoor stability | Outdoor | Indoor | Outdoor | Indoor |
| LOS/NLOS | LOS only | LOS/limited NLOS | LOS only | LOS/NLOS |
| Topology | P2P | P2P, limited P2MP | P2P | P2P, P2MP, MP2P, MP2MP |
| Mobility support | No | Possible | No | Yes |
| NLOS support | No | No | No | Yes |
| Transmitter | LD | LD/LED | LED | LD/LED |
| Receiver | PD | PD/Cam | Cam/IR | PD/Solar cell |
| Comm. Distance/range | >10,000 km | 20 m | 60 m | 10 m |
| Multi-user access | No | Yes | No | Yes |
| Interference | Yes | Yes | No | Yes |
| Outdoor vulnerable (Fog/rain) | Yes | Yes | Yes | Yes |
| Comm. direction | One direction | One direction | One direction | Bi-directional |
| Spectrum | IR/VL/UV | VL | IR/VL | IR/VL/UV |
| Cell size/service area | Small | Medium/small | Small | Ultra-small/small |
| Data rate | High speed 1–40 Gbps | LED = ~10 Gb LD = ~100 Gb | The achievable data rates are low~55 Mbps | LED = ~10 Gb LD = ~100 Gb |
| Security | High | High | High | High |
| Applications | It can be used to provide ultra-high speed backhaul connections within a data center | 1. Indoor 2. Backhaul 3. Vehicles 4. Outdoor 5. IoT 6. M2M | Applications on existing smartphones, such as: 1. Indoor positioning/navigation, 2. asset tracking and 3. the broadcast of barcodes 4. V2V 5. I2V | 1. Indoor 2. Outdoor 3. Underwater 4. Vehicles 5. IoT 6. D2D 7. M2M |
| 5G support | Yes | Yes | Yes | Yes |

**Table 5.** LiFi and Rf comparison [5,10,19–27,52,63,158,194].

| METRICS | LiFi | RF |
|---|---|---|
| Bandwidth | Unlimited | Limited |
| Power consuming | Medium | High |
| Topology | P2P, P2M | P2P, P2M |
| Data transmission carrier | Light waves | Radio waves |
| Communication medium | VL | RF |
| Range of spectrum | High | Low |
| Frequency | 100 TH$_z$ | 20 kHz–300 GHz |
| Range | ~10 m | ~100 m |
| Privacy | Higher | Lower |
| Underwater communication efficiency | High | Low |
| Dynamic environment support | Low to medium | High |
| Installation | Easy | Easy |
| Hybrid with other systems | Necessary | Not necessary |
| HO rate | High | Low |
| DL | Light waves | Radio waves |
| UL | RF, IR | Radio waves |
| Number of users per AP | Less | More |
| Video streaming | Very fast | Medium |
| IoT support | Yes | Yes |
| Transmitter | LED | Antenna |
| Receiver | PD, PV | Antenna |
| Standard | IEEE 802.15.7 | IEEE 802.11.xx |
| Cost | Cheap | Expensive |
| Availability | Where light exists | Limited |
| Environmental impact | Low | High |
| Services | Lighting and Internet access | Internet-only |
| Blockage | Yes | Limited |
| Interference | Low | High |
| Shadowing | Yes | No |
| Security | High | Low |
| Modulation | OFDM, OOK, CDMA, CSK, PM, etc. | DSSS, ASK, PM, etc. |
| Maximum data rate | 10 to 100 Gbps | 6 Gbps |
| Communication distance | Short | Long |
| Outdoor stability | Low | High |
| Indoor stability | High | High |
| Noise | Sunlights, Inter-Cell Interference ICI, and CCI | All electronic appliances |
| Health | Safe [7,52,63,194] | Harmful [5,194] |

## 8. Multi-User Access in LiFi

LiFi is differentiated from VLC technology when it comes to multi-point connections. LiFi supports multi-point connection provides multiple access to users or devices; several users may connect to the same AP. This feature is important, especially in a dynamic environment, such as hybrid networks. However, multiple IoT devices can be supported by LiFi networks [195]. Notably, LiFi supports user mobility and multi-user access. The central controller CU in LiFi also supports multi-user access [14]. The efficient implementation of a LiFi system is critical, so the appropriate multiple access technologies are also crucial. Many techniques for establishing multiple access will be discussed in this section. In LiFi communication, several multiple access techniques were implemented, which can be divided into two categories: (i) OMA and (ii) NOMA. In OMA schemes, users allocate different orthogonal time or frequency resources [195,196]. By contrast, NOMA allows multiple users to simultaneously utilising time and frequency resources through power domain Superposition Coded Modulation SPC and Successive Interference Cancellation SIC [197].

The performance gain of NOMA over OMA is mainly determined by the specific power allocation strategy adopted by the former [198]. Nevertheless, to apply NOMA in LiFi-enabled BID communication, the following two important issues [199,200] should

be considered: 1) Energy consumption: in LiFi-enabled BID IoT communication, energy consumption originates from the LiFi AP within each optical attocell and connected IoT devices. 2) Device QoS requirements: Devices can be divided into two categories; one includes low-speed devices, such as environmental sensors and health monitors; the other includes high-speed devices, such as multimedia-capable mobile phones. NOMA is another spectrum-efficient scheme that has been recently proposed for LiFi systems [197]. NOMA is based on multiplexing users in the power domain so that each user can utilize the full spectrum and time resources. The superior spectral efficiency enhancement offered by NOMA comes with an inevitable degradation in the corresponding error performance because of the reduction of the received SINR at the users' terminals [96,194,201]. An increase in the number of simultaneously connected users reduces the benefits of NOMA. More transmitted power is allocated for users with poor channel gains, whereas less is allocated for users with satisfactory channel gains [202]. NOMA shares the entire bandwidth among all simultaneous users, thereby allowing inter-user interference to occur, while OMA orthogonally allocates resources to all users.

To cope with such inter-user interference, SIC is applied on the receiving side to extract transmitted signals with superposition coding [203]. Study [203] proposed a hybrid, multiple access schemes that simultaneously uses NOMA and OMA in the same bandwidth. Moreover, the coexistence of OMA and NOMA in the entire bandwidth can reduce the actual number of simultaneous users in NOMA, thereby enhancing the effectiveness of NOMA. To improve the throughput of cell edge users, NOMA is used for RF communication systems. By utilizing the broadcasting nature of LEDs, the performance of a LiFi network can be enhanced efficiently with the application of NOMA [2].

Unlike conventional OMA technologies, NOMA can serve an increased number of users via non-orthogonal resource allocation, and it is considered a promising technology for 5G wireless communication [2]. Various multiplexing schemes of NOMA and OMA are presented in this paper. In [204], Carrier Sense Multiple Access (CSMA) was proposed. To avoid collisions, the LiFi attocell continuously senses the channel prior to transmission. Nevertheless, CSMA-based systems are highly susceptible to the hidden terminal situation, which results in significant degradation in system performance. Furthermore, CSMA involves the significant signalling overheads needed for request-to-send and clear-to-send symbols. The study [67] also found that Space Division Multiple Access SDMA-based LiFi attocells could simultaneously serve multiple users using the full spectrum and time resources. This was achieved by replacing the single-element transmitter with an angle diversity transmitter, which conveys separate beams to users in different locations. SDMA provides considerable spectral efficiency gains compared with TDMA. Still, it has the disadvantage of increased computational complexity because of the need for the careful design of the angle diversity transmitter and user-grouping algorithms. Moreover, SDMA fails to operate in scenarios where users are located at adjacent positions due to the high ICI.

In the study [205], Code Division Multiple Access CDMA-based systems and Optical orthogonal codes OOC are exploited to allow simultaneous network access for multiple users. The corresponding achievable spectral efficiency is limited by the large size and poor correlation characteristics of the OOC sequences. In the same context, Study [206], found that Orthogonal Frequency Division Multiple Access OFDMA could provide spectrum-efficient multiple access by sharing the available spectrum resources based on Orthogonal Frequency Division Multiple OFDM. In Study [207], NOMA was initially proposed as a multiuser access technique for RF cellular networks. Moreover, they discuss the use of DCO-OFDM in combination with NOMA for multiple users.

The study [96] presented the idea of Index-Time Division Multiple Access I-TDMA, a novel multiple access schemes that provided significant spectral efficiency enhancement compared with TDMA, which multiplexed users in the time domain like TDMA. The proposed I-TDMA harnesses the randomness of data sequences to create a new degree of freedom. This instance allows considerable spectral-efficiency gains compared with TDMA, with minimal added circuitry that maintains its low complexity compared with other

multiple-access schemes discussed earlier. An I-TDMA scheme can have minimal added complexity that involves using an XOR adder at the transmitting and receiving terminals. I-TDMA can be adapted to dynamically match the users' individual data-rate requirements by changing a single parameter [96]. Considering that no fast-fading characteristics are observed in the LiFi transmissions, high SNR statistics exist at low-frequency subcarriers. As a result, appropriate user-scheduling techniques are needed in OFDMA-based LiFi systems to ensure fairness among users. Moreover, the attained spectral gain comes at the expense of more complicated processing at the transmitter and receiver terminals.

SDMA shows promising performance in terms of RF, but it cannot be directly adopted in LiFi. One of the main issues refers to the transmitter. In RF, narrow directional beams are generated by changing the amplitude and phase of the signals transmitted by an antenna array. However, this approach cannot be implemented straightforwardly in LiFi because it uses Intensity Modulation IM and Direct Detection DD, and the main reason is that in LiFi, incoherent light sources are used, and thus the phase of the emitted waves cannot be fully controlled. However, in LiFi, LEDs have the inherent feature of a confined FOV. This characteristic can be exploited for generating directional light beams. Therefore, in optical SDMA, an angle diversity transmitter, which consists of multiple directional narrow FOV LED elements, is used as the optical transmitter.

By activating different transmitter elements, the angle diversity transmitter can serve multiple users at different locations simultaneously. An AP consists of a single-element transmitter that generates an omnidirectional signal. Only one user can be served within a time slot. Unlike TDMA, the SDMA AP consists of an antenna array that simultaneously generates multiple directional narrow beam signals. Hence, more than one active user can be served within a single time slot [67].

OFDM provides a straightforward method for multiuser access (i.e., OFDMA), where users are served and separated by a number of orthogonal subcarriers. However, no fast fading exists in IM/DD-based systems like LiFi systems; this signifies that low-frequency subcarriers have high signal-to-noise (SNR) statistics [140]; unlike RF systems, there is fast fading in wireless RF channels [63]. The indoor optical wireless channel shows a characteristic similar to the frequency response of a low-pass filter. Thus, subcarriers with lower frequencies generally provide users with high SNR statistics.

Therefore, using appropriate user-scheduling techniques is important in OFDMA to ensure that fairness in the allocation of resources (subcarriers) is maintained. OMA and NOMA schemes are summarized in Table 6.

**Table 6.** NOMA and OMA types with their characteristics [14,67,119,145,152,194–197,199–209].

| METRICS | OMA | | | | | | NOMA |
|---|---|---|---|---|---|---|---|
| | TDMA | I-TDMA | O-FDMA | SDMA | CDMA | CSMA | |
| System | LiFi | LiFi | LiFi | RF/LiFi | LiFi | WiFi/LiFi | LiFi, RF |
| Working principle | Time slot allocation | Time slot allocation | User scheduling; each sub-carrier is modulated independently and simultaneously | Separates signals by using different polarizations of the antennas. | Spreading digitalized analog signal over a wider bandwidth at a lower power level | Channel sensing | Users are allocated simultaneous time or frequency resources |
| Interference | Free | Free | High | Low | High | Low | Interference Cancellation using (SIC) |
| Spectral efficiency | Low | High | Medium | Medium | Limited | Hight | Highest |

**Table 6.** *Cont.*

| METRICS | OMA | | | | | | NOMA |
|---|---|---|---|---|---|---|---|
| | TDMA | I-TDMA | O-FDMA | SDMA | CDMA | CSMA | |
| Benefits | Guaranteed channel access | I-TDMA can be dynamically adapted to match the users' data rate requirements by changing a single parameter | Allows many low-bandwidth streams to transmit in parallel to reduce latency and jitter. Reduced latency is important for video and factory automation applications. | Reduced hidden node collisions | Robust against jamming and interference | The high packet success rate in dense traffic and distributed communications multi-service capabilities | Achieves superior spectral efficiency by serving multiple users simultaneously with the same frequency resource. Increases the number of users served simultaneously. |
| Complexity | Low | Low | Low | Medium | Low | Low, moderate | High |
| Transmission direction | UL/DL | UL/DL | UL/DL | UL/DL | UL/DL | UL/DL | UL/DL |
| System throughput | Low | Low | Low | Low | Low | Low | High |
| User-fairness | Low | Low | Low | Low | Low | Low | High |
| SINR at receiver | High | High | Normal | Normal | Normal | Normal | Reduced |
| Number of users | Less | Less | Less | Less | Less | Less | More users |
| Capacity | High | High | High | High | High | High | Low |
| Bandwidth | Orthogonally allocated | Orthogonally allocated | Orthogonally allocated | Orthogonally allocated | Orthogonally allocated | Orthogonally allocated | Shares the entire bandwidth simultaneously |

## 9. Open Issues and Discussion

In this section, the issues and challenges faced by the LiFi system are discussed according to our classification, specifically C1, C2, and C3. Issues and challenges in C4 are not considered in this section. HO, mobility, interference, and blockages are considered and analyzed. Some issues trigger another issue; for instance, the occurrence of blockage results in HO. For every problem, a specific method should be used. Figure 8 shows the causes and effects of the factors that are considered problems in the LiFi system.

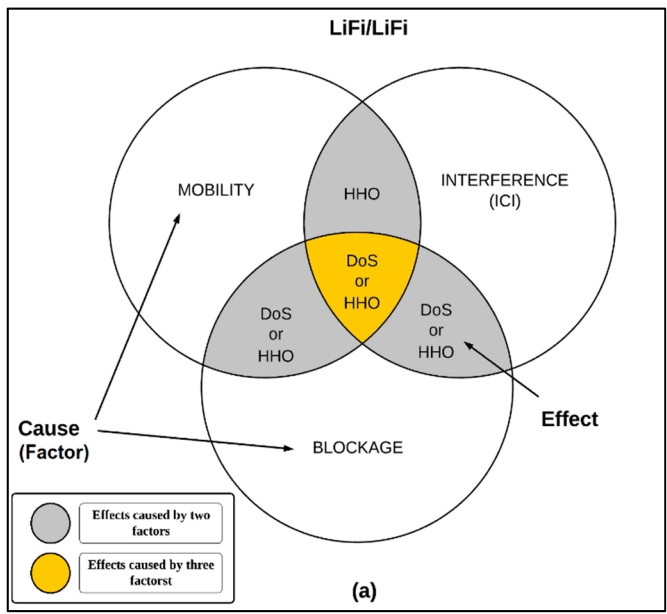

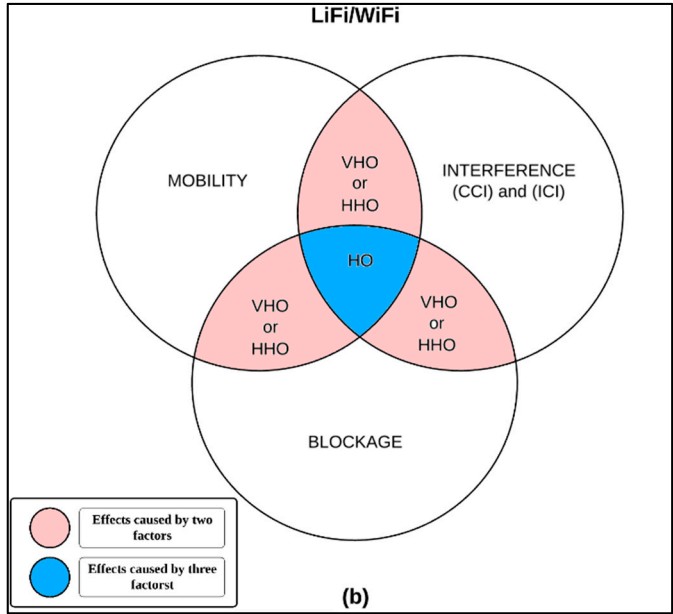

**Figure 8.** Relationship of issues in LiFi (**a**): LiFi/LiFi/(**b**) LiFi/Wi-Fi.

Given that the coverage areas of LiFi and WiFi completely overlap each other, a HO may occur. Hybrid LiFi/WiFi systems are susceptible to frequent HOs, HOs [108] between LiFi and WiFi VHO, and HOs between two LiFi AP Horizontal handover HHO. Study [108], proposed a HO scheme for a hybrid LiFi network. Study [40] suggested HO skipping to avoid frequent HOs. Notably, when users are moving, the HO rate would increase according to the speed of the movement.

The effect of HO on the system and users could be impactful and might lead to latency, the consumption of system resources, and UE. Study [66] considered the HO circle in their LB scheme by calculating the average HO efficiency and HO locations of moving users. The HO radius circle decreases when the system's throughput increases, and when WiFi throughput increases, the distance between HO locations and LiFi APs decreases. Study [31] proposed an adaptive HO scheme, where the HO rate was analyzed. However, the blockage was considered to have a relationship with HO.

In Figure 8, two cases are presented. The first case (Figure 8a) showing the issues that could be occurred in stand-alone LiFi network with multiple APs. The second case (Figure 8b) shows the issues in hybrid LiFi/Wi-Fi networks caused by similar factors. Various factors such as mobility, blockage, and interference can cause a particular effect on users served by the network. The effects can be VHO, HHO, and DoS may result from one or more factors simultaneously on connected users in both cases. For instance, user mobility and blockage could result in Denial-of-Service DoS or HHO for the connected user in the stand-alone LiFi network. However, in the hybrid system, no DoS occurs because the Wi-Fi signals are available as alternative connections when the LiFi user is blocked or during LiFi AP overload, which leads to VHO. The cause of a specific effect (issue) in the system. The colored area in the diagram represents the effect of multiple HO in (Figure 8b) could be one of both VHO and HHO. Each circle represents a factor or event which is the factors that occur to the connected user.

Figure 9 depicts a hybrid LiFi/WiFi network system reflected signal NLOS, blocked signal, and connected signal [57,82]. In fact, there are certain key drawbacks to the radio wave spectrum WiFi that encourage the use of LiFi and hybrid networks, such as (i) power, (ii) performance, (iii) accessibility, and (iv) security. Some problems and concerns should be resolved. First of all, LiFi needs LOS, which is critical to mobility and blockages. Thus, NLOS supports the connection but provides a much lower data rate [39,105].

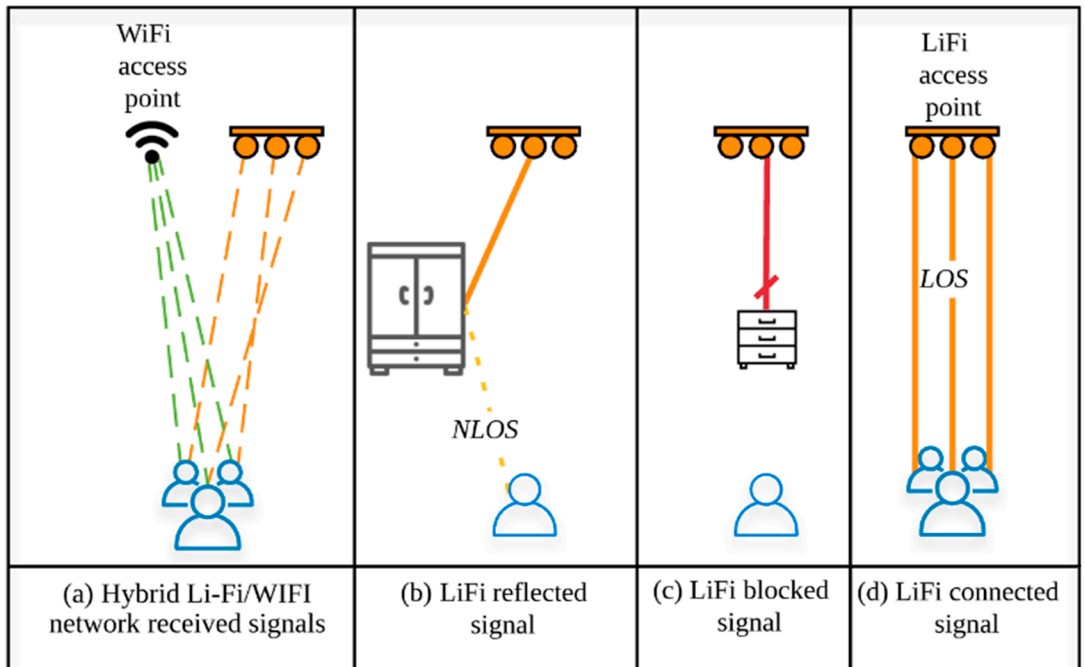

**Figure 9.** (**a**) Hybrid LiFi/Wi-Fi network, (**b**) reflected signal NLOS, (**c**) blocked signals by objects, and (**d**) connected LiFi LOS signals.

LiFi DLT does not work in the dark because light does not pass through artefacts; hence, the signal is automatically lost if the receiver is unintentionally blocked by a person walking in front of a light source [183,184]. The LiFi service area is small, and this issue has two possible solutions, precisely one for indoors and one for outdoors. The first one suggests the addition of APs to provide lighter and signal density. The second one indicates the deployment of more powerful light bulbs.

The first solution is suitable for indoor environments, such as houses, companies, industries, and inside vehicles, cars, and airplanes. The use of this method [180] would create many CCI between APs, which may add more challenges when applying the LB algorithm [66,79,82,84,94] HO management [66], and interference management and coordination [117]. The second solution is more suitable for outdoor places, such as fields,

open spaces, and streets, to provide a more prominent service area, a wide attocell, and a stronger optical signal that contributes to the optical gain [101]. However, this solution is inconvenient for indoor areas, where following international lighting standards is essential [210,211] because it may hurt the human eyes and lead to discomfort. Blockage and shadowing cause the non-uniform inclusion of the LiFi cells to further constrain the system throughput in the attocell [150].

LiFi service area, HO, and blockage are severe issues in LiFi networks. The causes and effects based on multiple criteria, such as blockage, shadowing, overhead area, interference, and network resources, are presented in this section. Note that blockage is different from shadowing. Specifically, blockage means the signals of a particular network are entirely blocked, thereby leading to the DoS in the stand-alone network and possibly to VHO in a hybrid network. Meanwhile, shadowing means less optical gain is received and may lead to HHO or VHO. When users cross the boundaries of a LiFi attocell service area, a HO and fluctuations in the network resources are triggered. In this analysis, the CCI is considered when interference occurs in two LiFi APs. In Table 7, all the causes and effects of these factors on the LiFi and hybrid LiFi/WiFi networks in multiple scenarios are presented: (i) 1 LiFi AP, (ii) 2 LiFi APs, (iii) 1 LiFi and 1 WiFi AP, and (iv) 2 LiFi and 1 WiFi APs. In this analysis, users' connection is assumed to be initially established on LiFi AP, where 1 means the connection is stable, and 0 means the user is disconnected from the AP. Lack of resources represents the low data rate received from the AP.

**Table 7.** Causes and effects of different metrics on LiFi, and LiFi/Wi-Fi hybrid networks that could trigger different types of events such as handover or even.

| Metrics | LiFi Stand-Alone Network | | Hybrid LiFi/WiFi Network | |
|---|---|---|---|---|
| | Scenario 1 | Scenario 2 | Scenario 3 | Scenario 4 |
| Blockage | 0 | HHO, 0 | VHO | HHO, VHO |
| Out of overhead | 0 | HHO, 0 | VHO | HHO, VHO |
| Shadowing | 0, 1 | HHO, 0, 1 | VHO, 1 | HHO, VHO, 1 |
| Noise/interference | 1 | HHO, 1 | 1 | HHO, VHO |
| Lack of resources | 1 | HHO, 1 | VHO, 1 | HHO, VHO, 1 |

## 10. Limitations and Future Directions

This study has reviewed, analyzed, and discussed in detail LiFi from various aspects in studies that have not been reviewed comprehensively. However, despite the strengths and contributions of this review, it still has several limitations that can be elaborated based on the following points. First, the papers downloaded and cited in this work were gathered from four main databases. Thus, future studies can include these four databases and may add more literature that they might find suitable in studying LiFi and communication. Also, other areas of focus within LiFi, including LiFi modulation with its types and usages across the literature, can be considered in future works in this domain. In addition, other related studies outside the eligibility criteria of this review (e.g., hardware equipment experiments in LiFi) can be investigated and reviewed in the future.

## 11. Conclusions

In this paper, we have extensively reviewed the LiFi technology from various aspects of the literature. The survey was extensively conducted on some of the major databases, and all eligibility criteria were applied to achieve the intended outcome of this review. The papers surveyed all the topics concerning LiFi technology. Based on this work, it has been distinctly identified that LiFi is an emerging field that fully warrants further investigation. This paper presented detailed information to be furnished for future researchers. These points have been summarized by analyzing and classifying the literature as a comprehensive review and by reviewing LiFi technology based on its essential aspects. This paper

addressed the main highlights: LiFi system and applications, components and architecture, the ULT and DLT, the attocell, and the service area. The advantages and disadvantages of LiFi and a comparison of LiFi with other technologies are also presented. Multi-user access techniques used in LiFi are explained. It is evident that most of these highlights and contributions require further investigations to provide timely future work opportunities for researchers interested in this interdisciplinary field.

**Funding:** The APC was funded by University Journal Publication Grant, Research Management Centre (RMC), University Putra Malaysia, Seri Kembangan, 43300, Selangor, Malaysia.

**Institutional Review Board Statement:** Not applicable.

**Informed Consent Statement:** Not applicable.

**Conflicts of Interest:** The authors declare no conflict of interest.

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
