# Peer review of "A Review on LiFi Network Research: Open Issues, Applications and Future Directions"

_applsci, doi:10.3390/app112311118_

Round 1
Reviewer 1 Report
This work summarizes LiFi technology quite well; however, there’s issues to deal with especially the definition and scope of LiFi compared to visible light communications.
- The concept ‘LiFi’ and ‘visible light communication (VLC)’ is interchangeably mix-used in many literatures. Some treat them exactly the same. Some other say LiFi includes more of networking concept. Or, some think VLC and LiFi have the relationships like RF communication and WiFi: the former being a more broad concept and the latter being more indoor application. You should first find and collate these different points of view. In short, the differentiation and explanation between LiFi and VLC in this work are insufficient and unclear. There should be clear comparisons and systematic comparative analysis on the two concepts (LiFi and VLC), before doing an in-depth review.
- The query you used is only about LiFi. There are much more literatures being published a year in the name of VLC, although the technologies are similar to those you summarized in the name of LiFi in this work. This means the statistics you show here are incomplete. Please search more with VLC.
- The concept and definition of modern VLC were first investigated in Japan, although the name ‘LiFi’ was coined later. Please include some of the Japanese frontiers in VLC as well.
Author Response
First of all, we would like to thank the “Nemanja Mancic” Special issue’s Editor the journal of “MDPI Applied Sciences” and the anonymous reviewers for their excellent suggestions and comments. Without their efforts, this manuscript would not have been in its current shape and form. Based on the valuable comments of the reviewers, we have thoroughly revised the manuscript. We hope that the reviewers will find that the current revision is up to their required standards. Below we detail our rebuttal.

Reviewer 2 Report
Dear authors, I have a few questions/suggestions regarding the reviewed paper:
- Why did you not consider HW based implementation studies? Why you chose to include only simulation studies are part of your review? In my opinion although it is important to verify basic characteristics and performance through simulation, it is highly more important to test the feasibility of such new systems using real PoC implementations in order to verify the simulation results with real life data.
- Table 5 refers to RF spectrum as 2.4 or 5 GHz. This is of course wrong, a much wider range of the spectrum is used for wireless communications from MHz up to tens of GHz. If the table refers to Wifi (it seems it is indeed just referring to Wifi) and not RF in general, please update the table of the second column.
- It is not clear what its depicted and how is modeled in Figure 8. Please explain further the color mapping and harmonize the color code between the 2 sub figures.
- Propose to remove the phrase “LiFi does not work in the dark” and replace it with “Lifi is not working when fully obstructed (when there is no LOS and NLOS paths)”. “In the dark” means absence of any type of light. When Lifi is active, then by definition there is light, even if not in the visible range.
Grammar/spelling Corrections:
- Figure 6: Recevier - > receiver , Receving -> Receiving , Dawnlink - > Downlink
- Table 20, page 20: Lightening - > lighting.
There are more, i did not document all errors, please make another pass and remedy simply spelling and grammar issues present.
Author Response
Based on the valuable comments of the reviewers, we have thoroughly revised the manuscript. We hope that reviewer 2 will find that the current revision is up to their required standards. Below we detail our rebuttal.

Reviewer 3 Report
Very good and detailed review paper!!
Congrats to the authors!!!
No additional upgrades is needed.
Author Response
Dear reviewer, we are grateful for your valuable feedback. We’ve also improved our manuscript for resubmission including language mistakes grammar and spelling.
We would like to thank the reviewer for pinpointing the language mistakes. We hope the revised version will be up to his/her expectation.
Round 2
Reviewer 1 Report
The authors have revised the manuscript and adequately addressed the reviewers' comments. The presentation and clarity of the article have been greatly improved. I am happy for this article to be published in its current form.